# The Use of the WOFOST Model to Simulate Water-Limited Yield of Early Potato Cultivars

**Bogdan Kulig [1], Barbara Skowera [2], Agnieszka Klimek-Kopyra [1,\*] , Stanisław Kołodziej [3] and Wiesław Grygierzec [4]**

[1] Department of Agroecology and Plant Production, University of Agriculture in Kraków, Al. Mickiewicza 21, 31-120 Kraków, Poland; bogdan.kulig@urk.edu.pl

[2] Department of Ecology, Climatology and Air Protection, Faculty of Environmental Engineering and Land Surveying, University of Agriculture in Kraków, Al. Mickiewicza 24/28, 30-059 Kraków, Poland; barbara.skowera@urk.edu.pl

[3] COBORU Research Centre for Cultivar Testing (COBORU) Kochcice, ul. Zamkowa 11, 42-713 Kochcice, Poland; zdoo.kochcice@coboru.pl

[4] Department of Mathematical Statistics and Social Policy, University of Agriculture in Kraków, Al. Mickiewicza 21, 31-120 Kraków, Poland; wieslaw.grygierzec@urk.edu.pl

\* Correspondence: agnieszka.klimek@urk.edu.pl

**Abstract:** In this work, an attempt was made to use the WOFOST (WOrld FOod Studies) model to simulate the potential and water-limited yield of early potato cultivars Lord and Denar. Data from cultivar experiments carried out at the Polish Research Centre for Cultivar Testing in 2004–2013 were used in the study. The Lord cultivar yielded 22.4–67.8 t fresh tuber weight per ha and 3.8–11.5 t ha$^{-1}$ dry tuber weight during the study period. The highest tuber yields (over 10 t ha$^{-1}$ dry weight) were obtained in 2009, 2011 and 2012, and the lowest in 2005 (3.8 t ha$^{-1}$) and 2006 (2.65 t ha$^{-1}$). The water-limited tuber yield simulated by WOFOST ranged from 3.6 to 10.9 t ha$^{-1}$ dry weight and was about 0.45 t ha$^{-1}$ higher on average than the actual yield. The planting period each year was between days 104 and 120 of the year, and harvesting took place between days 216 and 232. Water availability was a factor limiting the yield. The yield limited by water deficiency was 38.7% lower (irrespective of the cultivar) than the potential yield. The WOFOST model was sensitive to water deficiency, and the simulated (water-limited) yields were close to the actual yield or showed a clear downward trend indicating evident rainfall shortages in 2005 and 2006.

**Keywords:** WOFOST; potato; stages of development; meteorological conditions

## 1. Introduction

The distribution of precipitation and air temperatures during the growing season has a significant impact on crop yields [1–3]. High temperatures adversely affect plant production, resulting in shorter growing periods and shorter biomass accumulation periods [4–6]. These changes are expected to progress and may even accelerate in the future, exerting a potentially serious but very uncertain effect on crop production. Potato, one of the most important non-grain commodity crops worldwide [7], is sensitive to rainfall and temperature variability during the growing season [8]. The early potato type is the most vulnerable to water shortage, due to its short growing period [9]. Many authors stress that early cultivars of potato are most sensitive to rainfall deficiency during tuber formation and maturation [7,10]. A rainfall shortage can decrease yield by more than 50% [11]. At the start of the growing season (April–May), weather conditions also influence tuber yield. During this period, when potato plants still have a moderate demand for water, cool, wet weather negatively affects crop yield

and quality [12]. In Central and Western Europe, large fluctuations in yield are observed due to more frequent rainfall shortages during periods of higher demand for water. This results in an increase in prices on the European market. The average price of edible potatoes in the EU-4 market (excluding the UK) for the 2015/16 season was about 220 €/tonne, compared to 165 €/tonne the previous season and about 55 €/tonne 2 years earlier. According to Eurostat estimates, EU-28 potato production in 2016 was 56.9 million tonnes [13]. In the EU-5 (Germany, Belgium, France, the Netherlands and the United Kingdom), total potato production decreased by 0.5% relative to the previous year and amounted to 33.7 million tonnes, accounting for 59% of the EU harvest. The current (2019) edible potato crop in north-west Europe is forecast to be nearly 20% smaller than last year's due to the summer drought, with reduced tuber size and quality.

One solution to this problem is to choose cultivars that are less sensitive to water shortages. The WOFOST (WOrld FOod Studies) model is a useful tool to verify the production potential of cultivars depending on the weather. Among deterministic models, WOFOST is the most adequate prediction tool because it simulates plant production on three levels: potential, water-limited and nutrient-limited (NPK). The model is used in the EU as a component of a broader system of monitoring and forecasting agricultural crop yields [14,15].

The aim of the study was to assess the suitability of the WOFOST model for simulating the potential and water-limited yield of early potato cultivars Lord and Denar.

## 2. Materials and Methods

The WOFOST 7.1 model was calibrated and validated using data from a multi-year field experiment conducted in Poland, in the Opole Voivodeship, at the Stare Olesno experimental station belonging to the company Hodowla Ziemniaka Zamarte Sp. z o. o. Stare Olesno is located in the Woźniki-Wieluń Upland, in the north-west part of the Próg Woźnicki region, known as Garb Olesna (50°54′ N, 18°21′ E, 230 m a.s.l.). The very early potato cultivars Lord and Denar, preferred by consumers for their taste attributes, were grown in the experiment. The soil of the experimental field was classified as agricultural suitability complex 4 and valuation class 4. It was light soil with a pH of 6.6. Potatoes were grown after maize, with organic manure (5 kg N, 1 kg P, 5 kg K per 1 tonne of manure) applied in autumn in the amount of 25 t ha$^{-1}$. Mineral fertilization was 133 kg N, 74 kg $P_2O_5$, and 198 kg $K_2O$. The potato cultivars were planted in the second 10 days of April and harvested in the second 10 days of August.

Data from COBORU (Research Centre for Cultivar Testing) cultivar experiments from 2004–2013 were used in the study. The data pertained to (1) the length of four stages of development (planting to emergence, emergence to start of maturity, start of maturity to full maturity, and full maturity to harvest); (2) total and marketable yield (The marketable yield of tubers was determined after removing tubers with a diameter below 35 mm and defective tubers from the total yield); and (3) starch content. The average daily air temperatures and total amount of precipitation were used as well.

The numerical data were used to calculate the basic yield characteristics of the potato cultivars, i.e., average, highest and lowest yield, standard deviation and coefficient of variation of total yield, marketable yield, and starch yield. The field was rainfed, not irrigated. The extent to which water demands were met was also assessed on the basis of potato rainfall requirements according to [16], by calculating the differences between the total precipitation in successive 10-day periods of the potato growing season and its precipitation requirements. To assess the magnitude of deficiencies and surpluses of rainfall relative to the average rainfall in successive 10-day periods, standard deviations of the 10-day total precipitation in 2004–2013 were calculated. The next stage of the study involved determination of how yield and its components were correlated with the total precipitation and number of days with precipitation in the four stages of development. Correlation coefficients were calculated between the two precipitation characteristics, i.e., the total precipitation and number of days with precipitation in each stage of potato development, and the total, marketable and starch yields.

## 2.1. Weather Data

In the years 2004–2013, both temperature and total precipitation at the Stare Olesno experimental station were variable relative to the average values from 1981–2010 (Table 1). In 2005 and 2006, rainfall was lowest in the period from June to August, with slightly greater 10-day shortages observed in June, a month when early cultivars have greater water demands than medium and late varieties. The most 10-day periods in which rainfall far exceeded the needs of the cultivars were noted in 2010 and 2011 (Table 2). In 2004–2013, the average monthly air temperature in the months of early potato growth (April–July) was 0.2–0.4 °C higher than the average values from 1981–2010. The average rainfall totals were 90% and 85% of the average from 1981–2010 in April and July, and 101% and 111% in May and June. The lowest rainfall occurred in the period from May to July in 2005, 2006 and 2008. Significantly higher rainfall compared to the average from 1981–2010 was recorded in 2009, 2010 and 2013.

**Table 1.** Temperature and precipitation relative to 1981–2010.

| Year | Mean Temperature | | | | | | Precipitation | | | | | |
|---|---|---|---|---|---|---|---|---|---|---|---|---|
| | IV | V | VI | VII | VIII | IX | IV | V | VI | VII | VIII | IX |
| 1981–2010 | 7.8 | 13.2 | 16.0 | 18.1 | 17.5 | 12.9 | 40 | 76 | 75 | 83 | 75 | 54 |
| 2004 | 8.3 | 11.6 | 15.5 | 17.4 | 18.3 | 13.1 | 66.6 | 55 | 70 | 51 | 110 | 27 |
| 2005 | 8.5 | 13.1 | 15.6 | 18.8 | 16.2 | 14.2 | 25.3 | 94 | 35 | 88 | 101 | 23 |
| 2006 | 8.5 | 12.9 | 17.0 | 21.7 | 16.7 | 15.4 | 74.1 | 45 | 60 | 2 | 114 | 42 |
| 2007 | 9.0 | 14.6 | 18.2 | 18.4 | 17.8 | 11.8 | 1.9 | 70 | 137 | 94 | 31 | 63 |
| 2008 | 8.0 | 13.0 | 17.9 | 18.6 | 17.8 | 12.3 | 59.1 | 58 | 31 | 64 | 98 | 39 |
| 2009 | 11.1 | 12.9 | 15.1 | 18.9 | 18.0 | 14.5 | 1.2 | 56 | 151 | 111 | 32 | 15 |
| 2010 | 8.3 | 12.1 | 16.5 | 19.8 | 18.0 | 11.9 | 34.8 | 231 | 66 | 123 | 107 | 119 |
| 2011 | 10.4 | 13.5 | 17.2 | 17.1 | 18.1 | 14.7 | 19.3 | 48 | 72 | 134 | 60 | 10 |
| 2012 | 9.6 | 14.5 | 17.0 | 16.5 | 18.5 | 14.0 | 24.8 | 37 | 81 | 68 | 57 | 46 |
| 2013 | 8.1 | 13.8 | 17.0 | 19.2 | 17.9 | 11.9 | 40.6 | 121 | 130 | 52 | 48 | 83 |

**Table 2.** Rainfall demand in 10-day periods during growth of very early potato according to Dzieżyc et al. [16] and deficiencies (−) and surpluses (+) of rainfall.

| Month | April | May | | | June | | | July | |
|---|---|---|---|---|---|---|---|---|---|
| 10-Days Period | 3 | 1 | 2 | 3 | 1 | 2 | 3 | 1 | 2 |
| Water requirement (mm) | 16 | 18 | 20 | 21 | 24 | 29 | 30 | 31 | 26 |
| 2004 | 26 | −1 | 5 | −13 | 3 | 8 | −24 | −6 | −15 |
| 2005 | −11 | 40 | 1 | −6 | 0 | −22 | −26 | −19 | −20 |
| 2006 | −15 | −10 | 4 | −3 | −2 | −20 | −1 | −31 | −25 |
| 2007 | −15 | −5 | 8 | 9 | 0 | 5 | 50 | 22 | 1 |
| 2008 | 1 | −8 | 19 | −12 | −13 | −21 | −19 | −10 | 7 |
| 2009 | −15 | −15 | −11 | 23 | 11 | 62 | −7 | 41 | −10 |
| 2010 | −10 | 22 | 121 | 29 | −4 | 10 | −23 | −22 | 17 |
| 2011 | −13 | 4 | −12 | −4 | −10 | −27 | 27 | 13 | 11 |
| 2012 | −10 | 4 | −13 | −13 | −3 | 21 | −21 | −8 | 0 |
| 2013 | −8 | 18 | 2 | 43 | 64 | −29 | 12 | −27 | −1 |
| Standard deviation | 13 | 17 | 39 | 20 | 22 | 29 | 26 | 23 | 14 |

## 2.2. Characterization of the WOFOST Model

Weather data from 2004–2013 were obtained from the meteorological station in Stare Olesno and used for calibration and validation. The CLIMGEN model [17] was used to analyse weather data, and IRENE software was used to assess the model [18]. In the WOFOST 7.1 model, the original values of the parameter coefficients in the 'plant' data file were adjusted for local conditions. Coefficients were changed for the following parameters: RGRLAI = 0.01(maximum relative increase in LAI—ha ha$^{-1}$ d$^{-1}$), SLATB = 0.00, 0.025 (specific leaf area—ha kg$^{-1}$), AMAX = 0.00, 20.00 (maximum leaf

$CO_2$ assimilation rate as a function of the development stage of the crop—kg/ha/hr), KDIF = 0.0, 1.0 (extinction factor for diffuse visible light), SPAN = 32. (life span of leaves growing at 35 °C—days), TSUMEM (temperature sum from sowing to emergence—°C), TSUM1 = 160 (temperature sum from emergence to start of tuber growth—°C), and TSUM2 = 1300 (temperature sum from start of tuber growth to maturity—°C).

The performance of the model was evaluated using the following statistical indicators: maximum error (ME), root mean square error (RMSE), coefficient of residual mass (CRM), modelling efficiency (EF), and coefficient of determination (CD). Formulae for calculating these model parameters are described in detail by Kabat et al. [19].

## 3. Results

Potato yield varied between years and depended on the weather and the cultivar (Table 3). The Denar cultivar produced higher tuber yields (48.0 t ha$^{-1}$). The lowest total yield (13.1 t ha$^{-1}$) and marketable yield (10.3 t ha$^{-1}$) were obtained for the Denar cultivar in 2006. The highest total tuber yield (81.6 t ha$^{-1}$) in 2011 was attained by the Denar cultivar, which demonstrates the significant variation in yield (CV 43%) determined by water availability during the growth and development period (Table 2). This was confirmed by analysing the correlation between yield elements and the total precipitation and number of days with precipitation in various stages of development (Tables 4 and 5). Statistically significant coefficients of correlation for total, marketable and starch yields with total precipitation and the number of days with precipitation were obtained for the first two stages. In the first stage of development (from planting to emergence), the total precipitation and the frequency of days with precipitation significantly, but negatively affected the yield elements. The excess water for plants in this stage significantly decreased the yield. In the second and third stages of development, the positive correlations indicate that the potato plants were more sensitive to total precipitation and the number of days with precipitation in this later period than during the first stage of development. Relationships between yield elements and the total precipitation and number of days with precipitation were also determined for longer periods, i.e., from planting to the start of maturity (1–2), planting to full maturity (1–3), start of maturity to full maturity (2–3), and planting to harvest (1–4). In the case of correlation of yield with precipitation features in these periods, the highest correlation coefficient (r = 0.87) was obtained for all yield elements with the number of days with precipitation during the period from the start of maturity to full maturity (2–3). The appearance of late drought during the tubering stage had a greater effect on the yield, because the efficiency of physiological processes in plants reaches a maximum in this period. The lack of significant correlation between yield elements and the total precipitation during the longest period (growth stages 1–4) indicates that at full maturity the plants required less precipitation than in the previous, shorter stages of development (1–3 and 2–3).

**Table 3.** Statistical characterization of the fresh tuber yield (t ha$^{-1}$) of Lord and Denar potatoes (average from all years).

| Cultivar | Denar | | | Lord | | |
|---|---|---|---|---|---|---|
| Yield t ha$^{-1}$ | Total | Marketable | Starch | Total | Marketable | Starch |
| Mean | 48.0 | 45.2 | 6.12 | 44.7 | 42.2 | 5.7 |
| Max | 81.6 | 81.3 | 9.87 | 67.8 | 67.8 | 8.07 |
| Min | 13.1 | 10.3 | 1.64 | 15.6 | 11.5 | 2.21 |
| SD | 20.9 | 22.4 | 0.25 | 17.0 | 18.6 | 2.0 |
| CV% | 43 | 49 | 40 | 38 | 44 | 34 |

**Table 4.** Spearman correlation coefficients between yield elements and the total precipitation in various stages of potato development.

| Precipitation in Indicated Stages of Growth (R) | Yield | Marketable Yield | Starch Yield |
|---|---|---|---|
| * R (1) | −0.51 * | −0.50 * | −0.45 * |
| R (2) | 0.69 * | 0.69 * | 0.60 * |
| R (3) | 0.25 | 0.30 | 0.11 |
| R (4) | −0.14 | −0.15 | −0.38 |
| R (1–2) | 0.40 | 0.44 | 0.31 |
| R (1–3) | 0.39 | 0.46 * | 0.25 |
| R (1–4) | 0.37 | 0.42 | 0.21 |
| R (2–3) | 0.66 * | 0.69 * | 0.50 * |

*—significance of correlation ($\alpha$ = 0.05). Digits in brackets indicate stages: (1); planting to emergence (2); emergence to start of maturation, (3); start of maturation to full maturation, (4); full maturation to harvest.

**Table 5.** Spearman correlation coefficients between yield elements and the number of days with precipitation in various stages of potato development.

| Precipitation in Indicated Stages of Growth (R) | Yield | Marketable Yield | Starch Yield |
|---|---|---|---|
| * L (1) | −0.69 * | −0.64 * | −0.66 * |
| L (2) | 0.65 * | 0.60 * | 0.67 * |
| L (3) | 0.34 | 0.40 | 0.27 |
| L (4) | −0.04 | −0.01 | −0.30 |
| L (1–2) | 0.47 * | 0.47 * | 0.48 * |
| L(1–3) | 0.56 * | 0.62 * | 0.52 * |
| L (1–4) | 0.50 * | 0.54 * | 0.34 |
| L (2–3) | 0.87 * | 0.86 * | 0.85 * |

*—significance of correlation ($\alpha$ = 0.05). Digits in brackets indicate stages: (1); planting to emergence (2); emergence to start of maturation, (3); start of maturation to full maturation, (4); full maturation to harvest.

*3.1. Comparison of Simulated and Empirical Time of Onset of Main Stages of Development of Selected Early Potato Cultivars (Calibration)*

In the first stage of verification of the model, the accuracy of simulation of the time of planting, emergence and full maturity (harvest) was determined on the basis of water availability. The average planting dates of the two potato cultivars determined empirically differed from the simulated ones. However, the simulation error was low, amounting to 1.77 days for planting, 1.04 days for emergence, and 1.74 days for maturity. Errors could cause simulated yields to differ from experimental yields by ± 0.5 t ha$^{-1}$ on average. This is a relative deviation of 0.9% for the average tuber yield. The mean standard deviation (SD) between the time of experimental and simulated emergence and maturity for the two varieties was 3.36 and 0.37 days for the Denar cultivar and 2.91 and 0.77 days for Lord (Table 6).

**Table 6.** Comparison of simulated (Sim) and empirical (Emp) lengths of stages of plant development (days): tuber planting, emergency and maturity.

| Item | PLANTING | | EMERGENCE | | MATURITY | | PLANTING | | EMERGENCE | | MATURITY | |
|---|---|---|---|---|---|---|---|---|---|---|---|---|
| | Sim | Emp | Sim | Emp | Sim | Emp | Sim | Emp | Sim | Emp | Sim | Emp |
| | Denar | | | | | | Lord | | | | | |
| Mean | 106 | 112 | 134 | 136 | 228 | 225 | 106 | 112 | 134 | 136 | 228 | 226 |
| SE | - | 1.77 | 1.04 | 2.11 | 1.74 | 1.86 | - | 1.77 | 1.04 | 1.97 | 1.74 | 1.49 |
| Median | 106 | 112 | 133 | 135 | 227 | 227 | 106 | 112 | 133 | 135 | 227 | 227 |
| SD | - | 5.59 | 3.30 | 6.66 | 5.50 | 5.87 | - | 5.59 | 3.30 | 6.21 | 5.50 | 4.73 |
| Min | 106 | 104 | 130 | 126 | 220 | 215 | 106 | 104 | 130 | 126 | 220 | 216 |
| Max | 106 | 120 | 140 | 144 | 236 | 232 | 106 | 120 | 140 | 143 | 236 | 232 |
| CV% | - | 4.99 | 2.46 | 4.89 | 2.41 | 2.60 | - | 4.99 | 2.46 | 4.57 | 2.41 | 2.09 |

*3.2. Validation of Simulated, Measured and Potential Yield of Selected Cultivars*

Potential yields of tuber dry weight ranged between 11.0 and 14.0 t ha$^{-1}$ (Figure 1). They were much higher than the empirical (actual) yields (2.2–11.6 t ha$^{-1}$). The simulated water-limited yield ranged from 3.48 to 11.7 t ha$^{-1}$. The results indicate that water availability was a yield-limiting factor. The yield limited by water deficiency was 38.7% lower (irrespective of the cultivar) than the potential yield. According to the simulations, total precipitation did not meet the rainfall needs of the plants.

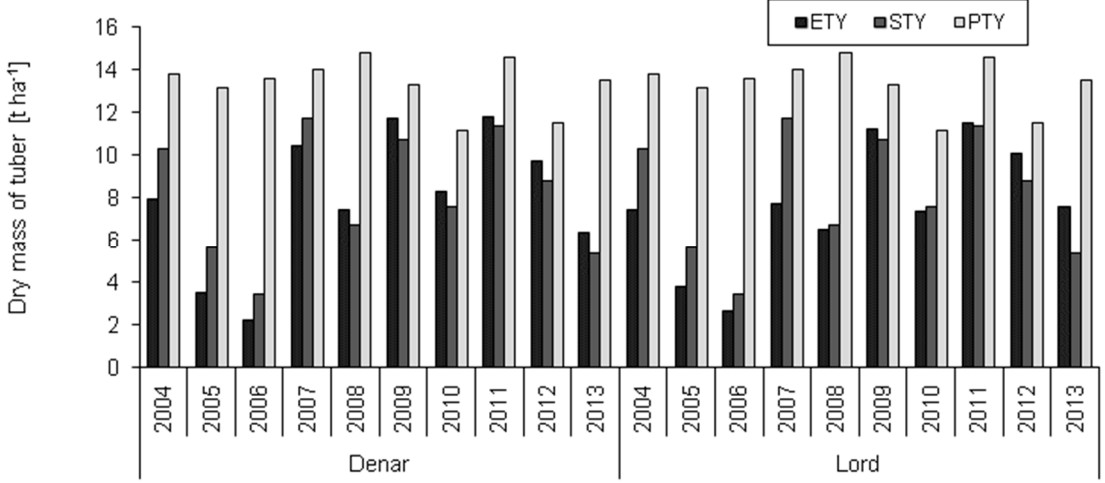

**Figure 1.** Calibration results of simulated, measured and potential yield of selected cultivars. (* dry matter), ETY—empirical tuber yield*, STY—simulated tuber yield*, PTY—potential tuber yield*.

The maximum error (ME) for tuber yield ranged from 2.3 t ha$^{-1}$ for the Lord cultivar to 3.9 t ha$^{-1}$ for Denar (Table 7). The root mean square error (RMSE) and the relative root mean square error (RRMSE) were used to evaluate prediction accuracy. The estimation errors for actual yield (RRMSE) for the equation were 17% and 24% for the Lord and Denar cultivars, respectively. The coefficient of residual mass (CRM), an indicator of overestimation or underestimation by the model, showed that WOFOST had overestimated the total tuber yield of the potato cultivars. However, the differences in CRM were minimal, and the values were very close to the ideal value of zero. The modelling efficiency value (EF) indicates the effectiveness of the model fit. Values below 0 indicate a worse fit of the model, compared to average measurement. The EF indicator was more accurate in the case of Lord (0.81) than Denar (0.52). The coefficient of determination (CD) indicates the ratio between the scatter of simulated values and the scatter of measured values. In the present study, its value was very close to the ideal value (1), as it did not exceed 1.2 for total and tuber dry matter yields of the two cultivars. The model performance was regarded as satisfactory, but was more accurate in the case of Denar than for Lord.

Yield simulations were burdened with an average error (RMSE) of 1.6 t ha$^{-1}$, and the coefficient of variation of the experimental yields was CV = 38% (Table 8). The mean bias error (MBE) between the simulated and experimental tuber yields was 0.4 t ha$^{-1}$.

There were statistically significant relationships between actual and simulated yields of the two potato cultivars, which are presented in the equations (Figure 2). The determination coefficient was 0.76 for the Denar cultivar and 0.52 for Lord. The equation indicates that the model overestimates the yield (Table 7).

**Table 7.** Statistical indicators of performance of the WOFOST model. ME = maximum error; RMSE = root mean square error; CRM = coefficient of residual mass; EF = modelling efficiency; CD = coefficient of determination.

| Atributes | Denar | Lord |
|---|---|---|
| RMSE | 1.8 | 1.3 |
| RRMSE | 24 | 17 |
| EF | 0.52 | 0.81 |
| ME | 3.9 | 2.35 |
| CRM | −0.08 | −0.03 |
| CD | 1.01 | 0.785 |

**Table 8.** Yield simulation errors (average for cultivars) for modified calibration of the WOFOST model. MBE—mean bias error; MAE—mean absolute error; RMSE—root mean square error; d—index of agreement.

| Parameters | *ETY* | *STY* | *PTY* |
|---|---|---|---|
| Mean | 7.79 | 8.19 | 13.4 |
| SE | 0.67 | 0.62 | 0.25 |
| Median | 7.68 | 8.22 | 13.6 |
| SD | 2.98 | 2.78 | 1.14 |
| Min | 2.23 | 3.48 | 11.2 |
| Max | 11.8 | 11.7 | 14.8 |
| CV% | 38 | 34 | 9 |
| MBE | −0.4 | | |
| MAE | 1.3 | | |
| RMSE | 1.63 | | |
| d | 0.91 | | |

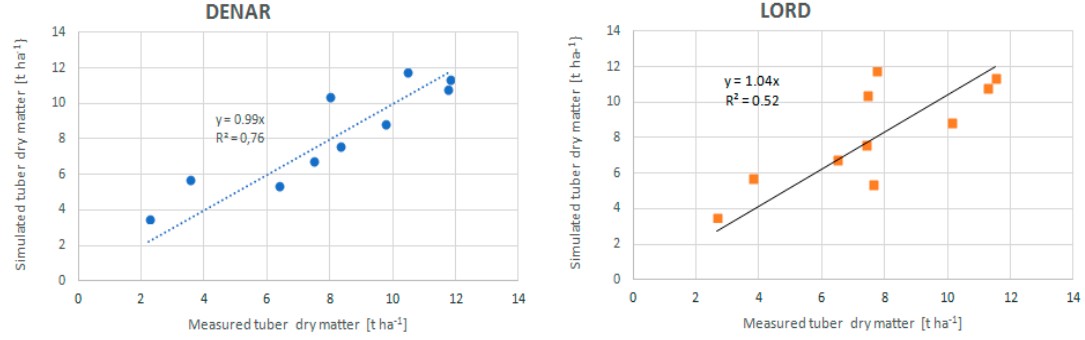

**Figure 2.** Relationship between simulated and actual yield of Denar and Lord potato.

## 4. Discussion

Potato cultivation in Central European climate conditions depends primarily on the volume and distribution of precipitation during growth [1]. Potato, due to its shallow root system, is considered to be the most drought-sensitive crop species [20,21]. Early and late varieties are particularly vulnerable to early stress, which is most detrimental to tuberization, bulking and tuber yield due to decreased leaf area, decreased photosynthetic rates and reduced partitioning of assimilates to tubers [22,23]. Because potato growth and tuber yield largely depend on rainfall, even a short period of water deficit can cause a substantial loss of tuber yield (50%) and deterioration of tuber quality [23–27]. In the climatic conditions of Southern Poland, where the total rainfall during the growing season is higher than in the regions where potato is grown in Central Poland, large fluctuations in yields are observed, which are more often caused by rainfall shortages during periods of higher water demand. Due to global

warming-related upward trends in air temperature during potato growth, while the rainfall regime remains unchanged, rainfall deficits contribute to poorer yields. Łabędzki and Bąk [26] emphasize that in periods with optimal air temperatures for potato, yield was determined by the distribution of total precipitation in successive months of potato growth. According to Van Loon [27], potato cultivation is vulnerable to water shortages due to the physiology of the plant. Potato has a high water content of about 85% dry weight, of which 1% is needed for metabolic processes and 99% for transpiration. Water stress can reduce photosynthesis efficiency at all stages of potato growth. A water shortage during the tuber-filling period causes the most significant crop loss, compared to drought at other stages of development. Our study showed that yield elements were associated with total precipitation and the number of days with precipitation over longer periods as well, but the period from the start of maturity to full maturity was the most effective for yield ($r = 0.87$). We assume that the appearance of late drought during the tuber bulking and tubering stages has a greater effect on yield because physiological processes in plants reach their maximum efficiency in this period. Haverkort and Goudriaan [28] have demonstrated that late droughts occurring during the tuber bulking stage have a greater effect on tuber yield due to increased crop transpiration, reduced formation of new leaves, and likely premature leaf shedding at this stage.

According to Rodriguez et al. [29], drought lasting up to 4 weeks during potato development results in a reduction in yield and deterioration of crop quality parameters. Although potato is very sensitive to water shortages, only precise irrigation (avoiding under- or over-irrigation) will result in high yield [30]. Otherwise, losses are observed not only in the tuber yield, but in economically valuable water as well, as demonstrated by El-Abedin et al. [11]. In the latest research, these authors have shown that a large amount of the water applied in an arid environment may be lost due to soil evaporation, thus resulting in poor crop performance and water productivity.

In the research results presented in this paper, the WOFOST model simulated the planting date at day 106 of the year and harvest between days 209 and 232. The entire period of observed plant development was between 106 and 123 days, while in the case of simulation the range was wider (103–126 days). The WOFOST model simulated potential production at the level of yield in the most favourable conditions, but in most cases the program overestimated the water-limited yield. The Lord cultivar yielded 22.4–67.8 t fresh tuber weight per ha and 3.8–11.5 t ha$^{-1}$ dry tuber weight during the study period. The highest tuber yields (over 10 t DW/ha) were obtained in 2009, 2011 and 2012, and the lowest in 2005 (3.81 t ha$^{-1}$) and 2006 (2.65 t ha$^{-1}$). The water-limited tuber yield simulated by WOFOST ranged from 3.61 to 10.9 t ha$^{-1}$ dry weight and was on average about 0.45 t ha$^{-1}$ higher than the actual yield. Reidsma et al. [31], in a simulation of water-limited potato yield, noted 23% lower yield than potential yield. In our research, the water-limited yield was 39% lower (irrespective of the cultivar) than the potential yield, and at the same time water availability was shown to be a factor limiting early potato yield. The WOFOST model was very sensitive to water deficiency, so simulated yields were close to the actual yields or showed a clear downward trend, indicating evident rainfall shortages in certain years of the study.

## 5. Conclusions

1.  The model adapted to the agro-climatic conditions of central Europe enabled the simulation of early potato yields with a relative error of RRMSE = 20%. The empirical and simulated data were better matched in the case of the Denar variety ($R^2 = 0.76$) than for Lord. Average yields from all years and the simulated yield were similar, with an MBE (mean bias error) of only 404 kg (in favour of yields simulated by WOFOST).
2.  Water availability (SSMB) was a yield-limiting factor. The water-limited yield was 38.72% lower (irrespective of the cultivar) than the potential yield. WOFOST is very sensitive to water deficits, and simulated water-limited yields were close to the actual ones or showed a clear downward trend, indicating evident rainfall deficits in 2005 and 2006.

3.　In the climatic conditions of Central Europe, where precipitation deficits are increasingly observed. The WOFOST model seems to be a suitable tool for forecasting the yield of early potato.

**Author Contributions:** S.K. was responsible for funding acquisition and the design of the experiment, while B.K., W.G. were responsible for data simulation in the WOFOST model and statistical analyses. A.K.-K., B.S. wrote the first draft of the article and all authors reviewed, edited and accepted the final manuscript. All authors have read and agreed to the published version of the manuscript.

**Funding:** This work was supported by the Ministry of Science and Higher Education of the Republic of Poland.

**Conflicts of Interest:** The authors declare no conflict of interest.

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
