# Peer review of "The Use of the WOFOST Model to Simulate Water-Limited Yield of Early Potato Cultivars"

_agronomy, doi:10.3390/agronomy10010081_

Round 1
Reviewer 1 Report
The manuscript “The use of the WOFOST model to simulate water-limited yield of early potato cultivars” (ID: agronomy-625054) is aimed to provide an interesting tool for predicting the magnitude of potato yield reduction as a function of limited water availability to plants.
The work has been appropriately developed, though the authors should address the following comments, also highlighted in the attached pdf revised version.
Introduction
Line 32: the citation 5 should be replaced with the one suggested in the pdf revised manuscript attached.
L 35: the citation suggested in the pdf revised manuscript should be added.
Materials and Method
L 75 and 86: is it ‘precipitation total’ or ‘total precipitation’?
Results
The unit of measurements used across the text, tables and figures should be uniformed and, in this respect, I recommend to use ‘t ha-1.
L 159-160: in Table 6 some integrations are needed, as highlighted in the pdf revised manuscript.
L 187-188: in Figure 2 some expressions should be as apex format.
Discussion
L 200-205: these sentences should be modified, as they have already reported in the Introduction section.
References
The formatting of all the citations should be checked and done in compliance with this Journal style, as highlighted in the pdf revised manuscript attached.
Dear Authors, you should address all my recommendations highlighted across the text, tables and figures.

Author Response
Thank you for revision of our manuscript.
The all changes are visible in text.
Line 32: the citation 5 should be replaced with the one suggested in the pdf revised manuscript attached.
The citation 5 was replaced, according to Rev1 suggestion.
L 35: the citation suggested in the pdf revised manuscript should be added.
The citation was added, according to Rev1 suggestion.
Materials and Method
L 75 and 86: is it ‘precipitation total’ or ‘total precipitation’?
It was changed to “total precipitation’
Results
The unit of measurements used across the text, tables and figures should be uniformed and, in this respect, I recommend to use ‘t ha-1.
The units were uniformed to ‘tha-1’
L 159-160: in Table 6 some integrations are needed, as highlighted in the pdf revised manuscript.
The Table 6 was integrated.
L 187-188: in Figure 2 some expressions should be as apex format.
It was corrected
Discussion
L 200-205: these sentences should be modified, as they have already reported in the Introduction section.
The sentence were modified.
References
The formatting of all the citations should be checked and done in compliance with this Journal style, as highlighted in the pdf revised manuscript attached.
The all citations were checked and corrected.

Reviewer 2 Report
Review of: the use of WOFOST model to simulate water limited yield of early potato cultivars
General comments
Overall I think the article is well structured and flows well. The results look nice.
I was only confused by table 8.
I also did miss some information in the methods and materials mainly concerning the field trails and WOFOST. Information on the version of WOFOST used as well as the values for the altered parameters would be useful in reviewing the accuracy of the results.
Within some of the tables in the results values are mentioned, that do not seem to be mentioned in the text.
There were also a number of grammatical errors, mainly in the beginning of the paper.
Specific comments
The abstract could do with a mention of the location of the fields and the years of the field trails. I think that it should also be mentioned in the materials and methods that the locations of the fields are in Poland. While the region names and coordinates are nice, it required a google search for me to find the location.
Within this article different units are used to describe the yield or yield related stuff. For example, in line 45 for the price of potatos euro/tonne is used. Then in line 119 the (fresh) yield is mentioned in dt/ha. In line 155 the yield difference is written down in t ha- 1. Later on in line 165 kg ha-1 is used for the (dry) weight/yield. Lastly in line 229 t.ha-1 is used. It is easier to compare numbers if the same units are used. And for consistency I would stick to one layout format i.e. t/ha or t ha-1.
I’m not familiar with the soil description used in line 67.
When reading the methods and materials it took me a while to realize that these fields were rainfed and not irrigated. A short mention early in the materials and methods could clear this up.
On lines 67 till 70 you talk about the fertilization. I was wondering what manure was applied and what the NKP content of this manure was.
I did not see any mention of (ir)radiation data required for WOFOST. Did you require this data or was this not needed for the version of WOFOST used?
In part 2.2 (lines 103 till 116) you mention the calibration of WOFOST and the parameters altered. I think this would be a great place to mention the version of WOFOST used. Also, while it is useful that you mention the parameters altered I would also like to know to what you altered these parameters.
In the results on lines 133 to 136 you talk about the correlation of yield and precipitation in the different growth stages. You mention that the highest correlation coefficient was found between start of maturation till full maturity, this would be stage 3. However, in table 5 this seems to be the lowest correlation coefficient.
In Tables 3,4 ,5 etc. you mention yield, trade yield and starch yield. While you do refer to all of them in tables 4 and 5 this is not the case for table 3. Even when you mention them you only mention them shortly. I think it would be interesting data to explore further.
I was wondering why the period between full maturity and harvest was included. I would not expect any yield increase in this period since there are no aboveground parts of the plant in this period. Why was stage 4 included? This also makes me wonder about the significant result for R(1-3) and not significant difference for R(1-4) in trade yield in table 4.
Table 7 includes EF which is not mentioned within the text.
In line 182 you mention the upper limit of CD. However, based on the value for Lord (0.785) I would be more interested in if this value meets a lower limit.
The Parameters in Table 8 are not explained in the text. I’m unfamiliar with these specific parameters. Since they are not explained in the text I did not understand this table to the text accompanying it.
In the discussion in line 218 you mention the effect of drought. However, you don’ t mention an excess of water. Considering the large amount of rainfall in 2007,2009 and 2012 I would be interested to see if this played a role.
Grammatical
On line 20 it was written down that the planting period each year was between the 104th and the 120th. This should be 104th and 120th.
The sentence on lines 22 and 23 It was stated The yield limited by water deficiency... I do not believe this is grammatically correct.
Line 29 starts with The. This should be removed.
On line 35/36 it is stated that “The most expose to water shortage is early potato type,...” It seems that the you mean to say: The most sensitive to water shortage are early potato cultivars,....
In the next sentence (line 36/37) it is mentioned that both early and late cultivars are most sensitive to rainfall deficiency during tuber formation and maturation. Due to the way this sentence is structured it took me couple reading to realize that is was this period in the growing season that both early and late cultivars are most sensitive and not that both more sensitive than the other.
Layout
Something went wrong with the layout of table 6. Instead of centered Lord is right next to the right edge of the page.
In Figure 1 the Y-axis states tuber yield. While I can see by the amount and text prior that you refer to the dry yield I would add this into the Y-axis label as well.
Line 171 the order is STY ETY PTY while figure 1 has ETY STY PTY. I would stick to one order. That would make it easier to read.
In line 242 you mention that Denar is better. Nowhere in the conclusion do you mention the cultivar Lord. Since this is the conclusion I would suggest to add better than Lord.
On line 246 you mention WOFOST reacted very sensitively to... This is grammatically in correct. Either WOFOST reacted strongly to... or WOFST is very sensitive to...
Author Response
Respond to Review 2
I would like to thank for all comments and suggestion. The all changes were inserted in to the text of manuscript and are visible.
In abstract section whole shortcomings were corrected.
In methodology section it was added the missing information about experimeny, location, version of WOFOST.
The outcomes presented in text were exposure in text.
The units were uniform in whole text.
The information related to model were added.
The grammatical errors were corrected.
The layout of tables was improved.
Specific comments

Reviewer 3 Report
This manuscript contains useful and interesting results on potato modeling and should be published after revision. Most of my comments are just about wording.
[1]Line 16, round numbers to nearest 0.1
[2]Same for lines 18 and 19
[3]Line 22, round percentage to 39%
[4]Line 23, delete “highly”
[5]Line 26, delete “yield”
[6]Line 31-32, change to “and shorter biomass accumulation periods”
[7]Line 33, delete “the”
[8]Line 34, add comma after “crops”
[9]Line 34, change to “non-grain crops worldwide [7], sensitive to . . .[8]. The most vulnerable to”
[10]Line 37 to 38, I don’t understand why early are late cultivars are most sensitive.
[11]Line 38, change to “can decrease yield more than”
[12]Line 49, change to “59%”
[13]Line 49, what is the “eating potato crop”?
[14]Lines 74 and elsewhere, What is the difference between “total yield” and “commercial yield”? Define the differences the first time they are used.
[15]Line 89 change to “both temperature and “
[16]Line 90, “highly dynamic”: I am not sure what that means. Might consider using better wording here.
[17]Line 95, delete “about”
[18]Lines 128 to 130, I don’t understand this statement.
[19]Lines 134 to 136, Was this expected? Explain why this occurred.
[20]Table 4, why commas used with the numbers?
[21]Table 5, why commas used?
[22]Line 151, “accuracy of simulation of time of planting”: Based on what? Temperature?
[23]Lines 153 to 158, errors of 3 days or less are very low, in my opinion. Might emphasize that. Also, round these errors to the nearest whole number.
[24]Table 6, why the commas?
[25]Table 7 ME, round to nearest whole numbers.
[26]Line 175 and elsewhere, why use dt/ha some places and kg/ha elsewhere? Be consistent throughout.
[27]Line 178, statement starting with “However, the differences”, Why are 2355 and 4000 closed to zero?
[28]Line 196 round these two percentage numbers to the nearest whole number.
[29]Figure 2, round the r2 values to the nearest 0.01.
[30]Sentence on line 215 starting with “A water storage”, this explains the correlation during the maturity stages, right. Need to emphasize that.
[31]Line 234 round to nearest whole number.
[32]Same for line 245.
Author Response
Thank you very much for revision of our paper.
The all suggestions and comments and incomprehension were applied are visible in text of manuscript.
In abstract section whole shortcomings were corrected.
The outcomes presented in text were exposure in text.
The units were uniform in whole text.
The grammatical errors were corrected.
The layout of tables was improved.
Specific comments were taken into consideration.
Round 2
Reviewer 1 Report
Dear Authors, you have addressed my recommendations and, therefore, the manuscript can be published in this Journal in my opinion.
Author Response
Thank you for your positive comments for our manuscript.
Reviewer 2 Report
General comments
I think the article is much improved. It reads much better, the tables are easier to understand and I found your discussion to be stronger.
However, I still have some questions regarding the model calibration. The description is slightly expanded but I still miss the values for the parameter coefficients that you used. Currently you included some of the original values for model parameters. Based on these values I assume you started with the plant data file potato 701? I would like to see the actual parameter coefficients that you used for your model runs that deviated from the original file.
Apart from this I spotted some grammatical errors. These should be easy to fix.
Specific comments
Overall I would advise checking for double spaces. I spotted a few of them.
Within the abstract I could spot an error in line 17: Data from polish... Polish could be capitalized and the correct form would be data from the Polish...
In line 41/42: A rainfall shortage can decrease yield more than 50%. Please change this to decrease yield by more than 50%.
In line 53 you mention edible potatoes. I do believe you actually mean ware potatoes or did I misunderstand this sentence?
In line 80: and total of precipitation should be and total amount of precipitation
In line 97: variablye
In line 137/138: In the second and third stages of development, the positive correlations between traits were presented. It indicates that the potato Take out the in the positive correlations and change It to This in the last sentence.
In line 185 you forgot to change the , into a . in the numbers
In Table 6 there are some bold numbers, why are these bold?
There seems to be a change in line distance between the paragraph ending on line 272 and the next.
Author Response
We sent manuscript to correction and inserted all sugestion
